# Bayesian-LoRA: LoRA based Parameter Efficient Fine-Tuning using Optimal Quantization levels and Rank Values trough Differentiable Bayesian Gates

**Cristian Meo** [* 1]  **Ksenia Sycheva** [* 1]  **Anirudh Goyal** [2]  **Justin Dauwels** [1]

## Abstract

It is a common practice in natural language processing to pre-train a single model on a general domain and then fine-tune it for downstream tasks. However, when it comes to Large Language Models, fine-tuning the entire model can be computationally expensive, resulting in very intensive energy consumption. As a result, several Parameter Efficient Fine-Tuning (PEFT) approaches were recently proposed. One of the most popular approaches is low-rank adaptation (LoRA), where the key insight is decomposing the update weights of the pre-trained model into two low-rank matrices. However, the proposed approaches either use the same rank value across all different weight matrices, which has been shown to be a sub-optimal choice, or do not use any quantization technique, one of the most important factors when it comes to a model's energy consumption. In this work, we propose Bayesian-LoRA which approaches low-rank adaptation and quantization from a Bayesian perspective by employing a prior distribution on both quantization levels and rank values. As a result, B-LoRA is able to fine-tune a pre-trained model on a specific downstream task, finding the optimal rank values and quantization levels for every low-rank matrix. We validate the proposed model by fine-tuning a pre-trained DeBERTaV3 on the GLUE benchmark. Moreover, we compare it to relevant baselines and present both qualitative and quantitative results, showing how the proposed approach is able to learn optimal-rank quantized matrices. B-LoRA performs on par with or better than the baselines while reducing the total number of bit operations by roughly 70% compared to the baseline methods.

[1]Delft University of Technology, NL. [2]Google Deep-Mind, UK. *Equal Contribution. Correspondence to: Cristian Meo <c.meo@tudelft.nl>, Ksenia Sycheva <K.Sycheva@student.tudelft.nl>.

Accepted to the Workshop on Advancing Neural Network Training at International Conference on Machine Learning (WANT@ICML 2024).

## 1. Introduction

Pre-trained language models (PLMs) have become the de-facto models in various natural language processing tasks (Devlin et al., 2019; Liu et al., 2019; He et al., 2021b; Radford et al., 2019; Brown et al., 2020b). Although full fine-tuning (FT) has been the most common way to adapt pre-trained models to downstream tasks Qiu et al. (2020); Raffel et al. (2020), with the rise of large pre-trained models full FT is becoming unfeasible. For instance, while BERT (Devlin et al., 2019) consisted of up to 300 M parameters, GPT-3 (Brown et al., 2020b) contains up to 175 B parameters, making full FT extremely computationally and energy demanding. The main lines of research to address this issue focus on reducing the fine-tuning parameters while maintaining or even improving the downstream performance of PLMs. One approach is to mitigate such a problem by adapting only some parameters or learning external modules for new tasks, while keeping the base model frozen and shared across tasks. As a result, only a small number of task-specific parameters need to be stored and loaded, greatly boosting the operational efficiency when deployed. For example, Adapter Tuning approaches (Houlsby et al., 2019; Rebuffi et al., 2017; Pfeiffer et al., 2020; He et al., 2022) employ small neural modules called adapters within the layers of the pre-trained model. Prefix tuning (Li & Liang, 2021) and Prompt tuning (Lester et al., 2021) attach additional trainable prefix tokens to the input or hidden layers of the base model. These methods have been shown to achieve comparable performance to full fine-tuning, while only updating less than 1% of the original model parameters, significantly releasing the memory consumption.

Another Parameter Efficient Fine-Tuning (PEFT) line of research proposes to model the incremental update of the pre-trained weights in a parameter-efficient way, without modifying the model architecture (Zaken et al., 2021; Guo et al., 2020; Hu et al., 2022; Zhang et al., 2023; Valipour et al., 2022). Among this family of methods, the most widely used is LoRA (Hu et al., 2022), which parameterizes weight updates $\Delta$ as a low-rank matrix by the product of two much smaller matrices:

$$W = W_0 + \Delta = W_0 + BA, \tag{1}$$

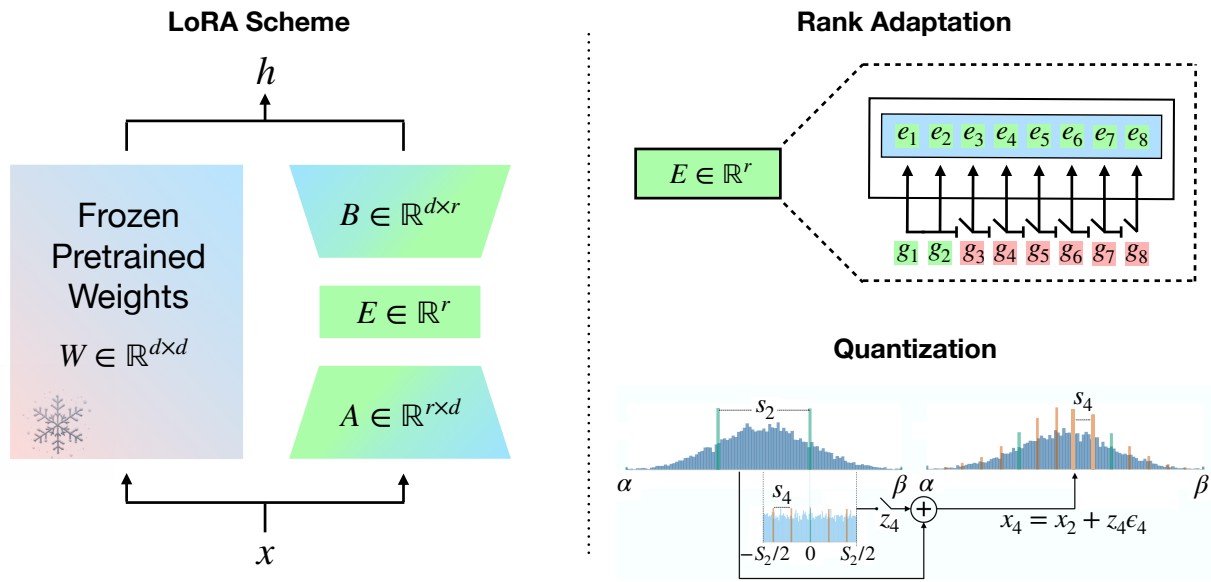

*Figure 1.* (Left) B-LoRA Scheme: As mentioned in Sec. 1, every weight $W$ can be decomposed as $W = W_0 + BEA$. (Right) Rank Adaptation and Quantization techniques are visually represented, following equation 13 for Rank Adaption and equations 7 and 8 for Quantization, respectively. Visual Representation of quantization technique is taken from (Van Baalen et al., 2020).

where $W_0, \Delta \in \mathbb{R}^{d \times d}$, $A \in \mathbb{R}^{r \times d}$ and $B \in \mathbb{R}^{d \times r}$ with $r \ll d$. During fine-tuning, only $A$ and $B$ are updated. The rank $r$ is chosen to be much smaller than the dimension of $W$ (e.g., $r = 8$ when $d = 1024$). With less than $0.5\%$ additional trainable parameters, training overhead can be reduced up to $70\%$, achieving comparable or even better performance than full fine-tuning (Hu et al., 2022). However, LoRA still has limitations since searching the optimal rank value requires re-running the entire fine-tuning for each new value (Valipour et al., 2022) and it sets the same rank $r$ of each incremental matrix $\Delta$ across different LoRA blocks (Zhang et al., 2023). The latter, as pointed out by Zhang et al. (2023), does not take into account that the impact of the weight matrices on downstream performances varies significantly across modules and layers when fine-tuning pre-trained models.

While PEFT approaches are proved to be very successful in reducing the number of parameters needed for specific downstream tasks, the LoRA-based approaches, proposed in the literature, either use the same rank value across all different weight matrices or do not use any quantization technique. However, to reduce the computational cost of neural network inference and the related energy consumption, quantization and compression techniques are often applied before deploying a model in real life (Van Baalen et al., 2020; Xu et al., 2024). Indeed, the former reduces the bit width of weight and activation tensors by quantizing floating-point values onto a regular grid, allowing the use

of cheap integer arithmetic, while the latter aims to reduce the total number of multiply-accumulate (MAC) operations required (Kuzmin et al., 2019; Krishnamoorthi, 2018).

Recently, Van Baalen et al. (2020) proposed the Bayesian-Bits approach, which introduces a novel and hardware-friendly decomposition of the quantization operation and allows for adaptable and optimal quantization levels, resulting in optimal quantization levels and, therefore, lower model energy consumption. Inspired by BayesianBits (Van Baalen et al., 2020), we propose Bayesian-LoRA (B-LoRA)[1] which approaches LoRA matrix decomposition and quantization from a Bayesian perspective. Indeed, by positioning a prior distribution on both quantization levels and rank values of the low-rank matrices weights, the optimal rank values and quantization levels for each individual LoRA block are learned. We validate the proposed approach, using the GLUE (Wang et al., 2019) benchmark, and compare it with state-of-the-art baselines, such as LoRA (Hu et al., 2022), DyLoRA (Valipour et al., 2022), and AdaLoRA (Zhang et al., 2023). Moreover, we perform a qualitative analysis of quantization levels and rank values across the fine-tuned quantized LoRA blocks, which shows how B-LoRA is able to reduce the total amount of bit operations of roughly $70\%$, while performing on par or better than the related SOTA baselines.

---

[1]Github link to Bayesian-LoRA implementation: https://github.com/KseniaSycheva/Bayesian-Lora

## 2. Related Work

### 2.1. Transformer-based Language Model

Pre-trained language models have gained significant attention in the field of natural language processing (NLP), due to their impressive capabilities in language generation, in-context learning, world knowledge, and reasoning.

The GPT family, including GPT-3 (Brown et al., 2020a), ChatGPT (OpenAI, 2022), GPT-4 (OpenAI, 2023), and InstructGPT (Ouyang et al., 2022) are some of the representative works on autoregressive LLMs. A second family of language models are bi-directional models, like De-BERTa (He et al., 2021b), DeBERTa-v3 (He et al., 2021a), RoBERTa (Liu et al., 2019), T5 (Raffel et al., 2020). It is a common practice to train transformer models on Language Modelling or Masked Language Modelling task in an unsupervised manner, which does not require annotated data, and adapt it for multiple downstream applications. Such adaptation can be done via fine-tuning, which updates all parameters of a model (Hu et al., 2022). Since transformer models often have billions of parameters, computing gradient updates for the entire model can be infeasible without appropriate hardware. This gave motivation for research work on parameter-efficient variations of fine-tuning (Hu et al., 2022; Zaken et al., 2021).

**Low-Rank Adaptation**. LoRA (Hu et al., 2022) is an approach that allows training model for a downstream task while updating only a small subset of weights. It models incremental updates of the weights being fine-tuned as a product of two matrices that have much fewer parameters. This results in the following forward pass:

$$Wx = W_0x + \Delta x = W_0x + BAx \qquad (2)$$

where $W_0, \Delta \in \mathbb{R}^{d \times d}$, $A \in \mathbb{R}^{r \times d}$ and $B \in \mathbb{R}^{d \times r}$ with $r \ll d$. Typically, $A$ is initialized from a Gaussian distribution and all entries of $B$ are set to 0. In transformers, LoRA is usually applied to weights in attention modules. Most of the experiments described by Hu et al. (2022) used queries and values only. He et al. (2022) extend it to weight matrices of FFNs (i.e., $W_{f_1}$ and $W_{f_2}$), leading to performance improvement. Meanwhile, they propose a unified view of various efficient tuning methods, including adapter tuning, prefix tuning, and LoRA. While LoRA (Hu et al., 2022) requires an expensive hyperparameter search to find the optimal rank values, DyLoRA (Valipour et al., 2022) proposes to fine-tune the model's weights for multiple rank values simultaneously. Inspired by Nested Dropout (Rippel et al., 2014), Valipour et al. (2022) truncates matrices $A, B$ to $A_b \in \mathbb{R}^{b \times d}$ and $B_b \in \mathbb{R}^{d \times b}$, sampling different rank values $b$ per iteration. In contrast to DyLoRA, which aims to optimize matrices for as many ranks as possible, AdaLoRA (Zhang et al., 2023) searches for optimal rank values. Given parameter budget, it is allocated among weights

according to their importance score. They reparameterize LoRA modules using SVD decomposition and during training diagonal values can be truncated or returned. Recntly, it was proven that a nearly linear time approximation exists for LoRA (Hu et al., 2024).

**Quantization of LLMs.** Quantization is a compression technique that reduces the bit width of the parameters and/or activations of LLMs to improve their efficiency and scalability (Xiao et al., 2023; Dettmers et al., 2022; 2023). Existing methods mostly focused on preserving or restoring the accuracy of quantized LLMs during the inference stage (Zhu et al., 2023), where the key is to reduce the memory footprint and computational costs without re-training the LLMs. In the context of low-rank adaptation, QLoRA (Dettmers et al., 2023) uses a novel high-precision technique to quantize a pre-trained model to 4-bit, and adds a small set of learnable low-rank Adapter weights that are tuned by backpropagating gradients through the quantized weights. Moreover, QA-LoRA (Xu et al., 2024) quantizes the weights of the pre-trained language model during fine-tuning to reduce time and memory usage. However, both QLoRA and QA-LoRA use vanilla LoRA blocks, inheriting their limitations related to rank values. In this work, we jointly optimize quantization levels and rank values to reduce the complexity of the model, while fine-tuning LoRA blocks to achieve better downstream performances.

## 3. Method

Our method searches for optimal precision and rank allocation in transformer models. In this section, we discuss these components separately.

### 3.1. Learnable Quantization

Following BayesianBits (Van Baalen et al., 2020), for a given weight $x$ with values in the range $[\alpha, \beta]$ we apply uniform quantization with different bitwidth $b_n = n, n \in \mathcal{N}$, where $\mathcal{N} = \{2, 4, 8, 16, 32\}$. For bitwidth $b_n$, quantized weights are computed as:

$$x_q = s\lfloor x/s \rceil, \qquad s = \frac{\beta - \alpha}{2^{b_n} - 1}, \qquad (3)$$

where $s$ is the step size of the quantized value and $\lfloor \cdot \rceil$ represents the round-to-nearest-integer function. Van Baalen et al. (2020) derive an expression for a residual error between consecutive quantization levels, using bitwidth $b_n$ and $b_{n+1} = 2 * b_n$:

$$\epsilon_{b_{n+1}} = s_{b_{n+1}} \left\lfloor \frac{x - x_{b_n}}{s_{b_{n+1}}} \right\rceil, s_{b_{n+1}} = \frac{s_{b_n}}{2^b + 1} \qquad (4)$$

Given this expression, weight $x$ can be reconstructed from its quantized version by adding error terms:

$$x_q = x_2 + \epsilon_4 + \epsilon_8 + \epsilon_{16} + \epsilon_{32} \qquad (5)$$

**Algorithm 1** B-LoRA block. Individual quantizer module parameters $\phi$ are not indicated for the sake of clarity.

---

**Require:** Input $x$, rank $r$, pre-trained matrix $W \in \mathbb{R}^{d_1 \times d_2}$, LoRA matrices $A \in \mathbb{R}^{r \times d_2}$ and $B \in \mathbb{R}^{d_1 \times r}$, vector with diagonal entries $E \in \mathbb{R}^r$, rank distribution parameters $\xi_2 \dots \xi_r$, quantizers $Q_w, Q_a, Q_e, Q_b$, used for weight matrices, and $Q_A, Q_E, Q_{\text{out}}$, used for output variables.

```
# quantize all weights
```
$\bar{W}, \bar{A}, \bar{E}, \bar{B} = Q_w(W), Q_a(A), Q_e(E), Q_b(B)$
```
# compute rank gates
```
$g_1 = 1, g_2 = \left\lfloor \sigma(\xi_2) \right\rfloor, g_i = \left\lfloor \prod_{j=1}^{i} \sigma(\xi_j) \right\rfloor$
```
# apply gates on diagonal entries
```
$\bar{E}_i = \bar{E}_i * g_i$
```
# compute output
```
**return** $Q_{out}(\bar{W}x + \bar{B} \cdot Q_E(\bar{E} \cdot Q_A(\bar{A}x)))$

---

To make weight precision controllable, gating variables $z_i, i \in \{4, 8, 16, 32\}$ are introduced:

$$x_q = x_2 + z_4(\epsilon_4 + z_8(\epsilon_8 + z_{16}(\epsilon_{16} + z_{32}\epsilon_{32}))) \quad (6)$$

Reinterpreting the model from a Bayesian perspective, we can introduce a prior distribution on gates $z_i$. The prior can be described with the following equations:

$$p(z_m|z_n = 1) = \text{Bern}(e^{-\lambda}),$$
$$\{m, n | m = 2 \times n, n \in \mathcal{N} \setminus \{32\}\} \quad (7)$$

that represent consecutive active gates, and

$$p(z_m|z_n = 0) = \text{Bern}(0) = 0,$$
$$\{m, n | m = 2 \times n, n \in \mathcal{N} \setminus \{2, 32\}\} \quad (8)$$

which are used for inactive gates. Notably, using this notation, whenever gate $n$ is inactive, all the consecutive ones will be inactive as well. Then, we can define the posterior distribution of gates $q_\phi$ as:

$$q_\phi(z_m|z_n = 1) = \text{Bern}(\sigma(\phi_m))$$
$$q_\phi(z_m|z_n = 0) = \text{Bern}(0) \quad (9)$$

where $\phi_i$ are used to parameterize the defined Bernoulli distributions and $\sigma(\cdot)$ is a sigmoid function. Van Baalen et al. (2020) provide results for convolutional models like LeNet (Simonyan & Zisserman, 2014) and VGG (Lecun et al., 1998). In our work, we apply learnable quantization to transformers. We limit our experiments by applying the method discussed above only to attention modules.

Consider an attention module, parameterized by matrices $W_k, W_q, W_v$ corresponding to keys, queries, and values, respectively. Following Van Baalen et al. (2020), we apply the learnable quantization approach to both weights

**Algorithm 2** Quantizer Module (Q); Hyperparameters $\zeta_1, \zeta_2$ and $t$ are fixed and defined in Appendix B

---

**Require:** Input $x$; Quantizer parameters $\phi$
clip(x, $\min = \alpha, \max = \beta$)
$s_2 \leftarrow \frac{\beta - \alpha}{2^2 - 1}, \quad x_2 \leftarrow s_2 \lfloor \frac{x}{s_2} \rceil$
$x_q \leftarrow x_2$
**for** $b$ in $\{4, 8, 16, 32\}$ **do**
  **if** training **then**
    $u \sim U[0, 1], \quad g \leftarrow \log \frac{u}{1-u}, \quad s \leftarrow \sigma((g + \phi)/b)$
    $z_b \leftarrow \min(1, \max(0, s(\zeta_1 - \zeta_2) + \zeta_2))$
  **else**
    $z_b \leftarrow \mathbb{I}\left[\sigma\left(\beta \log\left(-\frac{\zeta_2}{\zeta_1}\right) - \phi\right) < t\right]$
  **end if**
  $s_b \leftarrow \frac{s_{b/2}}{2^{b/2}+1}$
  $\epsilon_b \leftarrow s_b \left\lfloor \frac{x - \left(x_2 + \sum_{j<b} \epsilon_j\right)}{s_b} \right\rceil$
  $x_q \leftarrow x_q + z_b \left(\prod_{j<b} z_j\right) \epsilon_b$
**end for**
**return** $x_q$

---

and variables defined within the attention module. During fine-tuning, we define $W_k, W_q, W_v$ as LoRA blocks and optimize quantization levels of each weight and variable within the attention module. Specifically, we use a different quantizer for every matrix of each LoRA block $W_0, A, B$, and the related output variables.

### 3.2. Bayesian Rank Adaptation

In this section, we formalize the LoRA parametrization as in Zhang et al. (2023) and apply the gating mechanism defined in equation 6 to optimize the rank value of each LoRA block. We follow Zhang et al. (2023) and extend LoRA parameterization to have an SVD structure. As a result, LoRA blocks are modified to include the diagonal matrix $E$. Following Zhang et al. (2023), we store diagonal entries in a vector, therefore $E \in \mathbb{R}^r$. Hence, the forward pass in equation 2 can be expressed as:

$$Wx = W_0x + BEAx \quad (10)$$

In order to control and optimize rank values during training, the entries of the vector $E$ are multiplied by gating variables as follows:

$$\hat{E} = \left(\begin{bmatrix} g_1 \\ g_1 \cdot g_2 \\ \vdots \\ g_1 \cdot g_2 \cdots g_N \end{bmatrix} \times \begin{bmatrix} e_1 \\ \vdots \\ e_n \end{bmatrix}\right) \quad (11)$$

As for $z_i$ priors defined in equations 7 and 8, we define the $g_i$ priors as follows:

$$
\begin{aligned}
p(g_{n+1}|g_n = 1) &= \text{Bern}(e^{-\lambda}), \\
\{n|n \in 1, 2, \cdots, r-1\}, & \\
p(g_1) &= \text{Bern}(1)
\end{aligned}
\tag{12}
$$

where $p(g_1)$ is always 1 because all LoRA matrices should have at least rank 1. Such parametrization ensures that every diagonal entry $e_j$ is inactive if $e_i, j > i$ is not active. Consistently to equation 9, we can model the posterior distribution of gates $r_\xi$ as:

$$
\begin{aligned}
r_\xi(g_i|g_{i-1} = 1) &= \text{Bern}(\sigma(\xi_i)), \\
r_\xi(g_i|g_{i-1} = 0) &= \text{Bern}(0), \\
r_\xi(g_1) &= \text{Bern}(1),
\end{aligned}
\tag{13}
$$

The pseudocode for our method is provided in Algorithm 1. An algorithm for a forward pass of weight and activation quantizers can be found in Algorithm 2.

### 3.3. Training

As LoRA (Hu et al., 2022), our proposed approach is agnostic to any training objective. Consistently to prior works (Hu et al., 2022; Valipour et al., 2022; Zhang et al., 2023), we focus on language modeling as our motivating use case.

Suppose we are given a pre-trained autoregressive language model $P_\Phi(y|x)$ parametrized by $\Phi$. Consider adapting this pre-trained model to a given downstream task, represented by a training dataset of context-target pairs: $\mathcal{Z} = \{(x_i, y_i)\}_{i=1,..,N}$, where both $x_i$ and $y_i$ are sequences of tokens.

Following Hu et al. (2022), we can define the LoRA objective function as:

$$
\mathcal{L}_{\text{LoRA}}(\Theta) = \sum_{(x,y) \in \mathcal{Z}} \sum_{t=1}^{|y|} \log\left(p_{\Phi_0 + \Delta\Phi(\Theta)}(y_t|x, y_{<t})\right), \tag{14}
$$

where $\Phi_0$ represents the initial set of parameters of the pre-trained model and $\Delta\Phi(\Theta)$ represents the set of LoRA parameters that are optimized during the fine-tuning.

In order to optimize the proposed B-LoRA blocks, we follow the optimization scheme defined by Van Baalen et al. (2020). Since the gating variables are sampled from Bernoulli distributions, we use an approximation of the KL divergence

term, which results in the following objective:

$$
\mathcal{F}(\theta, \phi, \xi) = \mathcal{L}_{\text{LoRA}}(\Theta) - \lambda_q \underbrace{\sum_k \sum_{i \in B} \prod_{j \in B}^{j \leq i} q_\phi(z_{jk}|z_{ik} = 1)}_{\text{Quantization}} -
$$

$$
\lambda_r \underbrace{\sum_k \sum_{i=1}^r \prod_{j=1}^i r_\xi(g_{jk}|g_{ik} = 1)}_{\text{Rank Adaptation}}
\tag{15}
$$

where $B$ is a set of available bitwidth, $k$ denotes the index of the quantizer, $\lambda_q$ and $\lambda_r$ are hyperparameters that weight quantization and rank adaptation regularizers, respectively. In all our experiments, we set $\lambda_r = \lambda_q = 1$. We follow Van Baalen et al. (2020) and employ straight-through estimator (STE) (Bengio et al., 2013) for rounding operation, performing rounding in the forward pass, while using identity in the backward pass.

## 4. Experiments

In this section, we design empirical experiments to understand the performance of B-LoRA and its potential limitations by exploring the following questions: (1) How does optimizing quantization levels and rank values affects the downstream usefulness of LoRA-based fine-tuning approaches? (2) Can we observe consistent patterns of quantization levels and rank values across different tasks? (3) How many bit operations (BOPs) can we save by using adaptive quantization levels and rank values?

### 4.1. Experimental Setup

Following AdaLoRA (Zhang et al., 2023), B-LoRA is implemented for fine-tuning DeBERTaV3-base (He et al., 2020) on natural language understanding using the GLUE benchmark (Wang et al., 2018). We set the number of training epochs and scaling parameter alpha (Hu et al., 2022) according to AdaLoRA. However, while AdaLoRA uses specific hyperparameters for each different GLUE dataset, we use the same set for the whole benchmark, showing the robustness of the proposed method. Contrary to AdaLoRA, our method is applied to $W_k, W_q$, and $W_v$ while $W_o, W_{f_1}$ and $W_{f_2}$ are kept frozen. More details on hyperparameters are stated in Appendix B. The only layers that are fine-tuned with $W_q, W_k, W_v$ are two linear layers in the task-specific head. We provide the results for the full method B-LoRA(q + ra) and an ablation of it that uses only adaptive quantization B-LoRA(q). We can compute the number of training parameters for the proposed approach as follow:

$$
\#\text{params} = 6 \times r \times l \times d \tag{16}
$$

where $l$ represents the base model layers and $d$ is the hidden model's sizes, respectively. Number of parameters

*Table 1.* GLUE Benchmark. Here, $r$ parameter in LoRA and $b$ parameter in AdaLoRA correspond to rank value and parameter budget, respectively. We evaluate B-LoRA on two configuration: using quantization + rank adaptation (q+ ra) and using quantization only (q). Best results for each dataset are shown in **bold**, while second best ones are underlined. # of parameters refers to the number of trainable parameters of encoder (excluding classification head).

| Method | # Params | BOPs | MNLI Acc | SST-2 Acc | CoLA Acc | QQP Acc/F1 | QNLI Acc | RTE Acc | MRPC Acc | STS-B Corr |
|---|---|---|---|---|---|---|---|---|---|---|
| Full FT | 184M | | 90.12 | 95.63 | 69.19 | **92.40/89.80** | 94.03 | 83.75 | 89.46 | 91.60 |
| DyLoRA | 0.29M | 98.31 | 87.17 | 94.72 | 63.32 | 90.17 | 93.56 | 80.14 | - | 91.36 |
| LoRA (r=8) | 1.33M | 98.31 | 90.67 | 94.95 | 69.82 | 91.99/89.38 | 93.87 | 85.20 | 89.95 | 91.60 |
| AdaLoRA (b=576) | 1.99M | 95.32 | **90.77** | 96.10 | **71.45** | 92.23/89.74 | **94.55** | 88.09 | **90.69** | **91.84** |
| LoRA (r=2) | 0.33M | 97.44 | 90.34 | 94.95 | 68.71 | 91.61/88.91 | 94.03 | 85.56 | 89.71 | 91.68 |
| AdaLoRA (b=144) | 0.49M | 95.32 | 90.68 | 95.80 | 70.04 | 91.78/89.16 | 94.49 | 87.36 | 90.44 | 91.63 |
| B-LoRA (q) | 0.44M | **32.85** | 90.17 | **96.44** | 70.22 | 91.26/88.38 | 94.25 | 86.52 | 90.20 | 91.64 |
| B-LoRA (q + ra) | 0.44M | 32.91 | 90.27 | 96.33 | 69.63 | 90.75/87.79 | 94.2 | **88.33** | 90.03 | 91.76 |

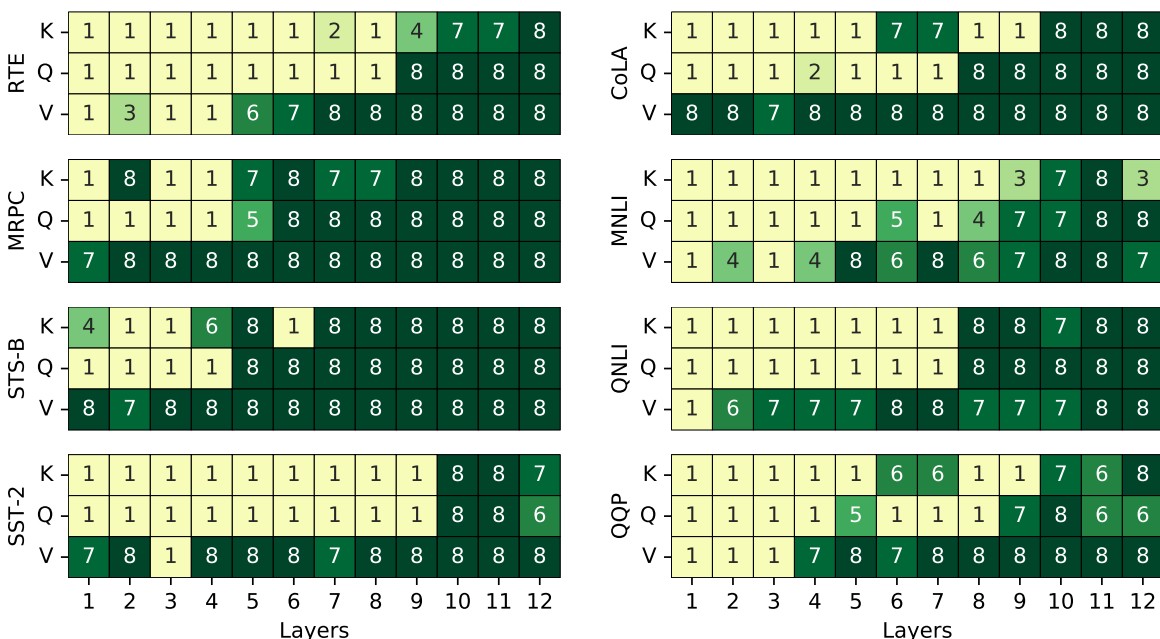

*Figure 2.* Rank distribution for GLUE benchmark. The last layers have larger rank values, compared to the first layers. Ranks of values $W_v$ are larger than ranks of keys $W_k$ and queries $W_q$.

in the classification head is not included in the parameter count, since it is fixed for all methods. A full description of B-LoRA and related baselines number of parameters computation can be found in Appendix D. B-LoRA is implemented using PyTorch (Paszke et al., 2019), publicly available HuggingFace Transformers weights (Wolf et al., 2019), BayesianBits[2] and AdaLoRA[3] repositories.

[2] https://github.com/Qualcomm-AI-research/BayesianBits
[3] https://github.com/QingruZhang/AdaLoRA/

**Baselines.** In order to assess the capabilities of the proposed method with respect to the current state of the art, we consider the following related baselines.

*Full Fine-tuning (FT)*: In this setup, the model is initialized with pre-trained weights and during training gradient updates are computed for all weights.

*LoRA* (Hu et al., 2022). It is a widely used method for parameter-efficient fine-tuning. Instead of fine-tuning the entire model, LoRA updates a subset of weights by representing the update matrices as a product of two matri-

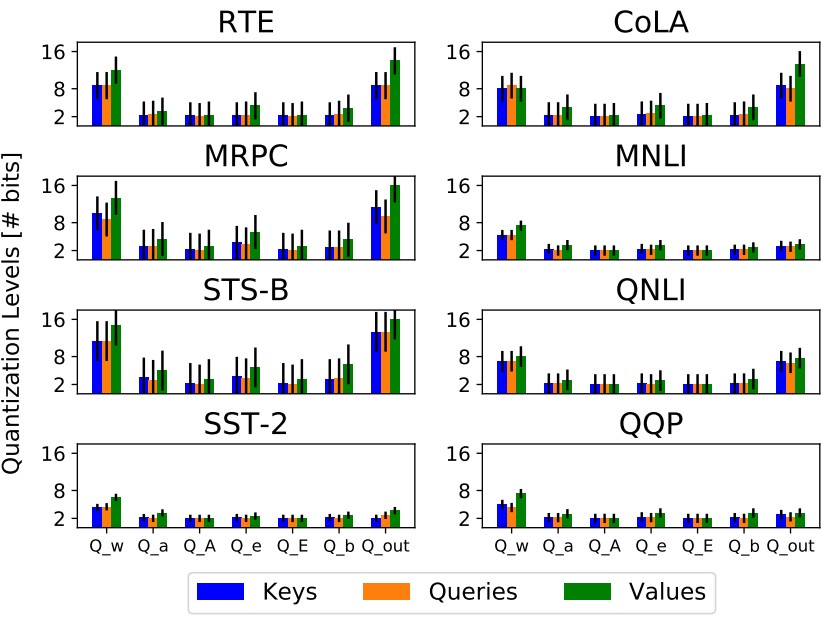

*Figure 3.* Quantization levels for GLUE benchmark. For each type of weight/activation, we compute median value of its bitwidth across the encoder. LoRA modules are kept in lower precision of 2, 4 bits. Values $W_v$ are kept in higher precision than keys $W_k$ and queries $W_q$.

ces with intrinsic dimensions much lower than the weight dimension, reducing the number of optimized parameters which can be controlled with intrinsic dimension. We use the same setup, used in (Zhang et al., 2023), for LoRA and AdaLoRA, which use DeBERTaV3 (He et al., 2021a) as pre-trained model and employ LoRA blocks in the following weights: $W_q, W_k, W_v, W_o, W_{f_1}, W_{f_2}$. We compute the number of parameters trained by LoRA as:

$$\#\text{params} = 2 \times r \times l \times (d \times 5 + d_i) \qquad (17)$$

where $d_i$ is the dimension related to the weight matrix $W_{f_1}$.

*AdaLoRA* (Zhang et al., 2023). It is an extension of LoRA that aims to limit the total sum of rank values used in different LoRA blocks. They define a computational budget and prune rank values according to an importance score (Zhang et al., 2023). We compute number of training parameters in AdaLoRA using Eq. 17 with $r$ which corresponds to the maximum rank value. According to Zhang et al. (2023), $r = \frac{b^T}{n}$ where $n$ is the number of adapted weights and $b^T$ is the target budget. We report the number of parameters for $b^T \in \{144, 576\}$, which results in $r \in \{3, 12\}$.

*DyLoRA* (Valipour et al., 2022): DyLoRA is another extension of LoRA, that enables adapting rank values dynamically. However, the goal of this method is to optimize the model fine-tuning for a range of ranks, in such a way that different versions of the fine-tuned model can be used if needed. Number of parameters for DyLoRA can be computed with Equation 17 with $r$ set to maximum rank.

**Metrics.** To assess the proposed approach and compare it to the related baselines we use two sets of metrics; downstream metrics, related to the GLUE (Wang et al., 2019) benchmark datasets, #params, and # Bit Operations (BOPs) to evaluate the efficiency of each method. Intuitively, the BOP count measures the number of multiplication operations multiplied by the bit width of the operands according to BOPs impact on the energy consumption of a model. To compute the BOP count we follow Van Baalen et al. (2020), which uses # Bit Operations as a hardware-agnostic proxy to model complexity and have an impact on energy level and device lifetime. According to Yang et al. (2017) and Van Baalen et al. (2020), BOPs impact the energy consumption of the deployed model. Moreover, Yang et al. (2017) points out how the number of bits accessed scales linearly with the corresponding bitwidth and that most of the energy is consumed by the multiplication operations, which scales linearly with the used variables bitwidth. Therefore, we use BOPs as a proxy measure to show how the proposed approach affects the energy consumption with respect to the related baselines. A list of the downstream metrics used for the GLUE benchmark can be found in Appendix E.

### 4.2. Results

**Quantitative Results.** Table 1 presents the comparison between the proposed model and the related baselines described in Section 4.1. On all datasets, B-LoRA achieves on-par performance with all other baselines, while presenting a much lower BOPs. Specifically, our method shows

slightly worse results for MNLI and QQP, but performs better than baselines on SST-2 and RTE (B-LoRA(q): 96.44 → AdaLoRA: 96.10 and B-LoRA(q+ra): 88.33 → AdaLoRA: 88.09, respectively). Interestingly, we can see that optimizing quantization levels and rank values results in better performances for RTE and STS-B datasets than using only quantization (B-LoRA(q+ra): 88.33 → B-LoRA(q): 86.52 and B-LoRA(q+ra): 91.76 → B-LoRA(q): 91.64, respectively). Moreover, Table 2, presented in Appendix A, reports B-LoRA BOPs for every dataset within the GLUE benchmark, showing how quantization levels and amount of BOPs are correlated.

**Qualitative Results: Task-Specific Head Quantization Levels.** We examine precision levels of task-specific head layers after fine-tuning. In all experiments layers of the task-specific head remained at the highest possible precision (32 bit). This result aligns with findings reported by Van Baalen et al. (2020), where they observed that the first and last layers were kept in higher precision in most of their experiments, however, we only observed higher precision in the last layers. Since Task-Specific Heads plays a central role when fine-tuning a pre-trained model, quantizying their weights has a big impact on downstream performances.

**LoRA blocks quantization levels and rank value patterns.** We analyzed the distribution of quantization levels and rank values after fine-tuning. Figure 3 shows that B-LoRA matrices are often kept with low precision of 2 or 4 bits, while pre-trained weights are usually kept with higher precision. A correlation between the quantization level of pre-trained weights and final output and the dataset size is present: the newer data the model observes during training, the less precision of pre-trained weights is needed. Indeed, datasets with a training set size below 10k (RTE, MRPC, STS-B, CoLA) present a median number of bits used above 8, while the remain ones (SST-2, MNLI, QNLI, QQP) use a median number of bits below 8. We hypothesized that there might be a correlation between specific attention weights (i.e., $W_k$, $W_q$, and $W_k$), optimal precision level, and related rank value. In accordance to our hypothesis, Figure 2 shows that $W_v$ has on average larger rank values, compared to $W_k$, $W_q$, which indicates that most of the information is retained within attention values. On the other hand, queries and keys can discard most of the information, since they are only used to compute attention weights and highlight the information retained within attention values. A similar pattern can be observed in Figure 3, where B-LoRA blocks used for values use more bits on average. In Appendix F, AdaLoRA rank values are provided for budget $b = 576$. The overall pattern observed in Zhang et al. (2023) aligns with our results, however, for B-LoRA rank reduction is more significant, since many LoRA modules are truncated to rank value 1.

## 5. Discussion

In this work we present B-LoRA, a parameter-efficient fine-tuning approach based on LoRA that allows to optimize quantization levels and rank values using Bayesian gating mechanisms proposed by Van Baalen et al. (2020). While works such as DyLoRA (Valipour et al., 2022) and AdaLoRA (Zhang et al., 2023) propose different approaches for optimizing rank values, they do not quantize variables and weights. Moreover, while our approach does not require any hyperparameter search, AdaLoRA requires specifying several hyperparameters for every dataset (i.e., computational budget, scheduler hyperparameters, learning rate). The main limitation of this work is that B-LoRA is only evaluated on the GLUE benchmark, while both LoRA and AdaLoRA provide results for natural language generation (Narayan et al., 2018; Hermann et al., 2015). In future works we will validate the model on the two question answering (QA) benchmarks SQuADv1.1 (Rajpurkar et al., 2016a) and SQuADv2.0 (Rajpurkar et al., 2018a), as well as the E2E benchmark (Novikova et al., 2017), using GPT-3 (Brown et al., 2020a) as pre-trained model.

## 6. Conclusion

In this study, we introduced Bayesian-LoRA (B-LoRA), a novel approach for optimizing quantization levels and rank values in model parameters, using Bayesian techniques. Our method extends the BayesianBits framework by Van Baalen et al. (2020), enabling a hardware-friendly and adaptive quantization that significantly reduces computational demands without sacrificing model performance. We empirically demonstrated that B-LoRA achieves competitive results on the GLUE benchmark, matching or even surpassing state-of-the-art methods such as LoRA, DyLoRA, and AdaLoRA, while also reducing bit operations by approximately 70%. This efficiency is achieved without the need for extensive hyperparameter tuning, contrasting sharply with approaches like AdaLoRA that require detailed configuration, tailored to each dataset. However, our evaluation was limited to the GLUE benchmark. Future work will aim to validate B-LoRA across a broader range of tasks, including question answering and natural language generation, using benchmarks like SQuAD v1.1 (Rajpurkar et al., 2016b) and 2.0 (Rajpurkar et al., 2018b), and the E2E generation benchmark (Novikova et al., 2017). Additionally, applying B-LoRA to other pre-trained models like GPT-3 (Brown et al., 2020a) will help establish its utility and robustness in diverse natural language processing contexts.

Overall, B-LoRA presents a promising direction for energy efficient, scalable, and effective model fine-tuning, making a step to bridge the gap between computational efficiency and performance.

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

## A. Additional Results

Table 2 illustrates how B-LoRA amount of BOPs varies across every GLUE dataset. As expected, datasets, showing the highest levels of quantizations, presented in Fig. 3, have the lowest amount of BOPs.

*Table 2.* GLUE Benchmark: BOPs. BOPs values for each dataset. Each value represents percentage w.r.t. BOPs of encoder and attention layers of LoRA with rank 16 applied on $W_q, W_k, W_v$ (BOPs of $LoRA_{r=16} = 100\%$, $LoRA_{r=2} = 97.04\%$), $AdaLoRA_{rmax=16} = 97.44\%$.

| | Relative BOPs in encoder | | | | | | | |
|---|---|---|---|---|---|---|---|---|
| Method | MNLI | SST-2 | CoLA | QQP | QNLI | RTE | MRPC | STS-B |
| B-LoRA (q) | 28.05 | 25.08 | 34.70 | 27.66 | 34.12 | 35.58 | 37.50 | 40.17 |
| B-LoRA (q + ra) | 26.67 | 24.38 | 34.19 | 25.04 | 30.87 | 35.21 | 36.99 | 42.08 |
| | Relative BOPs in Attention Layers | | | | | | | |
| B-LoRA (q) | 16.63 | 13.19 | 24.34 | 16.18 | 23.66 | 25.36 | 27.58 | 30.68 |
| B-LoRA (q + ra) | 15.48 | 12.84 | 24.15 | 13.60 | 20.32 | 25.32 | 27.32 | 33.24 |

## B. Training Details

In contrast to AdaLoRA, where different set of hyperparameters is used for every dataset as shown in Table 4, most of the hyperparameters in our experiments are the same for all datasets. The only value that is changed is number of training epochs, which can be found in Table 3. Table 5 reports hyperparameters used by DyLoRA and all hyperparameters that were fixed in B-LoRA experiments. Here $\zeta_1 \zeta_2$ are hyperparameters that ensure that $z$ has support for exact $0, 1$ and $t$ is a threshold used during inference for binarizing gates.

*Table 3.* Hyper-parameter setup of B-LoRA for GLUE benchmark.

| Dataset | # epochs |
|---|---|
| **MNLI** | 7 |
| **RTE** | 50 |
| **QNLI** | 5 |
| **MRPC** | 30 |
| **QQP** | 5 |
| **SST-2** | 24 |
| **CoLA** | 25 |
| **STS-B** | 25 |

*Table 4.* Hyper-parameter setup of AdaLoRA for GLUE benchmark. Reported from (Zhang et al., 2023).

| Dataset | learning rate | batch size | # epochs | $\gamma$ | $t_i$ | $\Delta_T$ | $t_f$ |
|---|---|---|---|---|---|---|---|
| **MNLI** | $5 \times 10^{-4}$ | 32 | 7 | 0.1 | 8000 | 100 | 50000 |
| **RTE** | $1.2 \times 10^{-3}$ | 32 | 50 | 0.3 | 600 | 1 | 1800 |
| **QNLI** | $1.2 \times 10^{-3}$ | 32 | 5 | 0.1 | 2000 | 100 | 8000 |
| **MRPC** | $1 \times 10^{-3}$ | 32 | 30 | 0.1 | 600 | 1 | 1800 |
| **QQP** | $5 \times 10^{-4}$ | 32 | 5 | 0.1 | 8000 | 100 | 25000 |
| **SST-2** | $8 \times 10^{-4}$ | 32 | 24 | 0.1 | 6000 | 100 | 22000 |
| **CoLA** | $5 \times 10^{-4}$ | 32 | 25 | 0.5 | 800 | 10 | 3500 |
| **STS-B** | $2.2 \times 10^{-3}$ | 32 | 25 | 0.1 | 800 | 10 | 2000 |

| Model | Parameter | Value |
|---|---|---|
| DeBERTa-Base-v3 | Optimizer | AdamW |
| | Warmup Ratio | 0.0 |
| | LR Scheduler | Linear |
| | Batch Size | 8 |
| | Learning Rate (LR) | 5e-4 |
| | Weight Decay | 0 |
| | LoRA Config | $r_q = r_v = r_k = 8$ |
| | LoRA $\alpha$ | 16 |
| | $\zeta_1$ | -0.1 |
| | $\zeta_2$ | 1.1 |
| | Threshold $t$ | 0.34 |
| | Max Sequence Length | 256 |
| | Seeds | 0, 1, 2 |
| RoBERTa-Base | Optimizer | AdamW |
| | Warmup Ratio | 0.06 |
| | LR Scheduler | Linear |
| | Batch Size | 32 |
| | Epochs | 30 |
| | Learning Rate (LR) | 4e-4 |
| | Weight Decay | 0.1 |
| | LoRA Config | $r_q = r_v = 8$ (unless otherwise mentioned) |
| | LoRA $\alpha$ | 16 |
| | Max Sequence Length | 512 |
| | Seeds | 10, 42, 4242, 10, 1010 |

*Table 5.* The hyperparameters that have been used in DyLoRA experiments with GPT-Medium and RoBERTa-Base and in B-LoRA experiments with DeBERTa-Base-v3.

## C. MACs and BOPs for LoRA

### C.1. MACs and BOPs

A MAC (Multiply-ACcumulate operation) is a multiplication followed by addition. This metric can be used to estimate complexity of the model and often dictate the memory usage of a network. It can be related to FLOPs as

$$\text{FLOPs} = 2 * \text{MACs}$$

MAC count of a common layers:

- linear: $\text{MACs}(l) = n_i * n_o$, where $n_i$ - number of input features, $n_o$ - number of output features

- convolution: $\text{MACs}(l) = C_o * W * H * W_i * W_f * H_f$, where $C_o$ - number of output channels, $W_i$ - number of input channels, $W, H$ - dimensions of output map, $W_f, H_f$ - dimensions of filter

A BOP corresponds to Bit OPerations, as defined in (Van Baalen et al., 2020). BOP count measures multiplication operations, multiplied by bit width of the corresponding components, which makes this metric a hardware-agnostic estimate of the complexity of a model. BOP count is computed the following way:

$$\text{BOPs}(l) = \text{MACs}(l) * b_w * b_a$$

where $b_w, b_a$ are weight and input activation bit width, respectively. BayesainLoRA method is additionally compared to AdaLoRA in terms of BOP count. Below derivation of BOP and MAC for self-attention mechanism is provided.

## C.2. Self-Attention MACs

Self-attention is a basic block of transformer models (Vaswani et al., 2017). For evaluating B-LoRA, BOP is computed for self-attention blocks of DeBERTa-v3 and compared to BOP of the same blocks with all weights and activation set to highest possible precision (32 bits).

Self-attention module is parameterized with 3 matrices $W_k, W_q, W_v \in \mathbb{R}^{\times}$ where $d$ is a hidden size of a model. Define maximum length of an input sequence as $l$, then

$$\text{MACs}(q) = \text{MACs}(k) = \text{MACs}(v) = d^2 * l$$

Other operation that increases MAC count for self-attention is dot product between keys and queries (attention scores). Assuming that number of attention heads is $h$, MACs of attention scores can be computed as

$$\text{MACs}(\text{attention\_scores}) = l^2 * \left\lceil \frac{d}{h} \right\rceil * h$$

Finally, values are weighted by attention probabilities, which gives

$$\text{MACs}(\text{attention\_scores}) = l^2 * \left\lceil \frac{d}{h} \right\rceil * h$$

Therefore, MAC count for a self-attention model can be computed as

$$\text{MACs}(\text{self\_attention}) = 3 * d^2 * l + 2 * l^2 * \left\lceil \frac{d}{h} \right\rceil * h + 1$$

where last term corresponds to a scaling factor.

## C.3. Disentangled Self-Attention MACs

Since in all experiments DeBERTa-v3 was used, MAC calculations need to be extended to attention variant proposed by (He et al., 2020). Disentangled attention utilizes positional information by introducing two extra matrices for keys and queries that are applied on positional embeddings. Then scores between positional keys and queries (context to position) and positional queries and keys (position to context) are computed and added to the attention scores.

Computations described above have components for which MAC need to be calculated. Assuming that positional embeddings size is $e$:

$$\text{MACs}(\text{pos}_k) = \text{MACs}(\text{pos}_q) = d^2 * e$$

For Context-to-Position and Position-to-Context dot product:

$$\text{MACs}(p_2 c) = \text{MACs}(c_2 p) = l * e * \left\lceil \frac{d}{h} \right\rceil * h$$

Each of them has a scaling factor. This results in

$$\text{MACs}(\text{dis\_self\_attention})$$
$$= \text{MACs}(\text{self\_attention}) + 2 * \text{MACs}(\text{pos}_k) + 2 * \text{MACs}(\text{p2c})$$
$$= 3 * d^2 * l + 2 * l^2 * \left\lceil \frac{d}{h} \right\rceil * h + 2 * d^2 * e + 2 * l * e * \left\lceil \frac{d}{h} \right\rceil * h + 3$$

## C.4. LoRA MACs

LoRA (Hu et al., 2022) parameterizes linear layer in the following way:

$$Wx = W_0 x + BAx$$

where $A \in \mathbb{R}^{\times}, B \in \mathbb{R}^{\times}$. MAC count for LoRA linear layer can be expressed as

$$\text{MACs}(\text{LoRA}) = \text{MACs}(\text{linear}) + (2 * r + 1) * d$$

## D. Number of Parameters

### D.1. LoRA

Number of parameters in one LoRA module with matrices $W \in \mathbb{R}^{d_1 \times d_2}$, $A \in \mathbb{R}^{r \times d_2}$, $B \in \mathbb{R}^{d_1 \times r}$ is computed with the following equation:

$$\#\text{params} = \#A + \#B = (r \times d_2) + (d_1 \times r) \tag{18}$$

LoRA is applied to 6 matrices in attention layer. $W_q, W_k, W_v, W_o$ have $d_1 = d_2 = d$, therefore, number of parameters in each of them is

$$(r \times d) + (d \times r) = 2 \times r \times d \tag{19}$$

Additionally, it is used in intermediate and output layers of attention, $W_{f_1} \in \mathbb{R}^{d \times d_i}$, $W_{f_2} \in \mathbb{R}^{d_i \times d}$. Number of trainable parameters in each of these layers is:

$$(r \times d) + (d_i \times r) \tag{20}$$

Summing parameters for all weights in attention layer results in:

$$4 \times (2 \times r \times d) + 2 \times ((r \times d) + (d_i \times r)) = 2 \times r \times (5 \times d + d_i) \tag{21}$$

For a model with $l$ layers, number of trainable parameters in the encoder is:

$$\#\text{params} = 2 \times l \times r \times (5 \times d + d_i) \tag{22}$$

### D.2. B-LoRA

B-LoRA is applied for $W_q, W_k, W_v \in \mathbb{R}^{d \times d}$. In total, it gives

$$\#\text{params} = 2 \times l \times r \times (3 \times d) = 6 \times l \times r \times d \tag{23}$$

parameters.

## E. GLUE Datasets Downstream Metrics

Table 6 provides details about GLUE datasets, such as task, number of examples in train/dev/test splits and metrics, used for evaluation.

## F. AdaLoRA Rank Distribution

Figure 4 shows the distribution of rank values in different layers in model, trained with AdaLoRA.

| Corpus | \|Train\| | \|Test\| | Task | Metrics | Domain |
|--------|-----------|----------|------|---------|--------|
| | | | Single-Sentence Tasks | | |
| CoLA | 8.5k | **1k** | acceptability | Matthews corr. | misc. |
| SST-2 | 67k | 1.8k | sentiment | acc. | movie reviews |
| | | | Similarity and Paraphrase Tasks | | |
| MRPC | 3.7k | 1.7k | paraphrase | acc./F1 | news |
| STS-B | 7k | 1.4k | sentence similarity | Pearson/Spearman corr. | misc. |
| QQP | 364k | **391k** | paraphrase | acc./F1 | social QA questions |
| | | | Inference Tasks | | |
| MNLI | 393k | **20k** | NLI | matched acc./mismatched acc. | misc. |
| QNLI | 108k | 5.7k | QA/NLI | acc. | Wikipedia |
| RTE | 2.5k | 3k | NLI | acc. | misc. |

*Table 6.* Task descriptions and statistics. All tasks are single sentence or sentence pair classification, except STS-B, which is a regression task. MNLI has three classes; all other classification tasks have two. Test sets, shown in bold, use labels that have never been made public in any form. Image is taken from Wang et al. (2019).

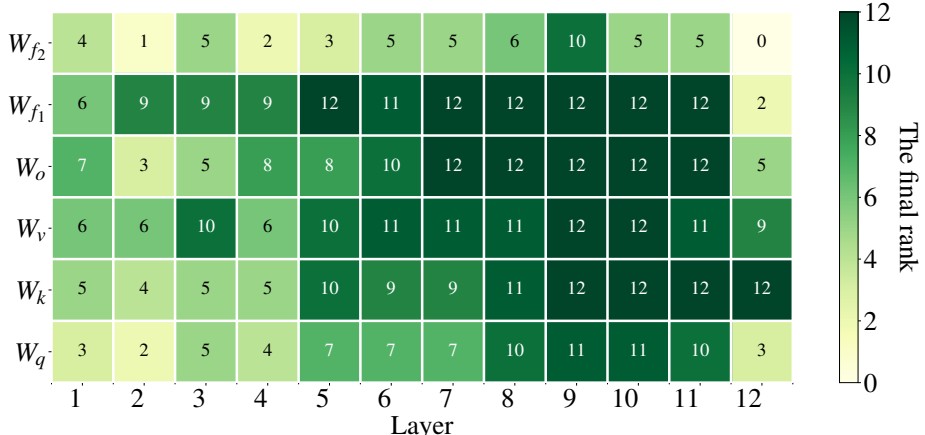

*Figure 4.* Rank Distribution for AdaLoRA on MNLI dataset.

