# OpenReview forum: "Bayesian-LoRA: LoRA based Parameter Efficient Fine-Tuning using Optimal Quantization levels and Rank Values trough Differentiable Bayesian Gates"
_ICML.cc/2024/Workshop/WANT — WANT@ICML 2024 Poster_

### Official Review · Reviewer_HMjZ · 2024-06-13
**Bayesian-LoRA is interesting but needs broader validation**

**Confidence:** 3

**Summary:**

The paper proposes Bayesian-LoRA (B-LoRA), a novel method for parameter-efficient fine-tuning of large language models. B-LoRA integrates Bayesian approaches for optimizing quantization levels and rank values during the fine-tuning process, aiming to reduce computational costs and energy consumption while maintaining or improving model performance.

**Strengths:**

B-LoRA introduces a novel integration of Bayesian methods for optimizing both quantization levels and rank values, providing a new dimension to parameter-efficient fine-tuning.

Unlike some existing methods, B-LoRA does not require extensive hyperparameter searches, making it more user-friendly and adaptable.

**Weaknesses:**

The paper does not clearly quantify the impact of B-LoRA on training time compared to other methods. The additional complexity of Bayesian optimization might increase training duration, which is a crucial factor for practical applications.

The evaluation primarily focuses on DeBERTaV3 and does not explore other pre-trained models like GPT-3, limiting the generalizability of the results.

---

### Official Review · Reviewer_3sHm · 2024-06-13
**Sound and instructive work; limited experiments.**

**Confidence:** 5

**Summary:**

The author used well-known bayesian tools and tricks in order to build adaptive quantization and low-rank model adaptation with adaptive rank. The work is easy to follow, clear, instructive.

**Strengths:**

+ Despite the fact the work gives new perspective on the old problem,
  cherry-picked results used in the paper are not novel at all. The work is
  comprised of multiple techniques proposed in prior works (quantization via
  Bernoulli RV and their practical parametrizations). However, I believe this
  the work is sound and instructive.

**Weaknesses:**

## Typesetting

+ Bibliography should be reviewed and actualized:  capitalization of titles,
  missing publication dates, journals conferences, etc (BitFit paper published
  at ACL conference https://aclanthohttps://aclanthology.org/2022.acl-short.1logy.org/2022.acl-short.1).
+ Missing table of content in hypertext markup.

## Major Points of Criticism

### Method

+ The bayesian perspective to LoRA is quite novel at the time. However,
  competitive works [1,2] has appeared recently.

[1]: https://openreview.net/forum?id=FJiUyzOF1m
[2]: https://openreview.net/forum?id=LZrCBQBCzl

### Experiments

+ It's great that the authors tried to make a holistic empirical study of
  proposed method and adopted experimentation protocol from (Hu et al, 2020).
  However, their comparison is not complete since their experiments are carried
  out only on DeBERTaV3-base. This fact limits contributions since experiments
  do not say anything about method scaling with model size increases. I believe
  that authors should experiment at least with DeBERTaV3-medium in order to
  fill the gap.

+ Lack of performance evaluation regarding training time is another
  flaw of the paper.

---

### Meta-Review · Area_Chair_bNek · 2024-06-15

**Recommendation:** Accept (Poster)
**Confidence:** 3

**Metareview:**

The paper revisits well-known Bayesian tools and tricks to build adaptive quantization and low-rank model adaptation with adaptive rank.

The reviewers acknowledge the technical novelty and insights of the current manuscript, but all (including the AC) criticize the soundness of the empirical evaluation. The AC would suggest including results on more tasks and neural architectures in the future manuscript.

---

### Decision · Program_Chairs · 2024-06-17

**Decision:**

Accept (Poster)

**Comment:**

We thank the authors for their time and contribution to WANT and we are pleased to share that after the reviewing process the paper has been accepted. Congratulations! We encourage the authors to consider reviewers' feedback for the improvement of the camera-ready version. We hope to see you in person at the workshop and brainstorm on efficient training research together!